# Gender Differences in Attachment Anxiety and Avoidance and Their Association with Psychotherapy Use—Examining Students from a German University

**DOI:** 10.3390/bs12070204

**Published:** 2022-06-22

**Authors:** Rainer Weber, Lukas Eggenberger, Christoph Stosch, Andreas Walther

**Affiliations:** 1Department of Psychosomatics and Psychotherapy, University of Cologne and Faculty of Medicine, University Hospital Cologne, 50931 Cologne, Germany; rainer.weber@uni-koeln.de; 2Department of Clinical Psychology and Psychotherapy, University of Zurich, 8050 Zurich, Switzerland; lukas.eggenberger@uzh.ch; 3Students Deans Office, University of Cologne and Faculty of Medicine, University Hospital Cologne, 50931 Cologne, Germany; c.stosch@uni-koeln.de

**Keywords:** psychotherapy use, attachment anxiety, attachment avoidance, gender differences, depression

## Abstract

Attachment anxiety and avoidance might explain gender differences in psychotherapy use, which is generally lower in men. In addition, university students are a particularly vulnerable group for mental health problems, and understanding psychotherapy use, especially among mentally distressed male students, is pivotal. A total of 4894 students completed an online survey answering questions regarding psychotherapy use and completing the PHQ-D identifying psychological syndromes. In addition, the ECR-RD12 was used to measure attachment anxiety and avoidance. Significant gender differences for attachment anxiety and avoidance emerged, showing higher attachment anxiety in female students and higher attachment avoidance in male students. Male students used psychotherapy significantly less than female students. Male students’ attachment anxiety and avoidance predicted psychotherapy use, while for female students, only attachment anxiety emerged as a significant predictor. Attachment anxiety is positively associated with psychotherapy use, and lower attachment anxiety in men may explain lower psychotherapy use in male students.

## 1. Introduction

Men and women differ with regard to the prevalence of several mental disorders with women showing higher rates in depressive disorders, somatoform disorders, anxiety disorders, and eating disorders [1,2,3,4,5,6], while men show higher rates in substance use disorders and especially in alcohol use disorders [7,8]. Furthermore, men show up to four-fold increased suicide rates and appear to be particularly vulnerable to commit suicide when facing economic or social losses [9,10]. This is in line with reports suggesting that overall men and women experience similar levels of psychological distress and mental disorders [11], yet men use psychotherapy about 30% less than women [12,13,14,15,16]. Whether this pattern differs with regard to differing psychological syndromes is, however, insufficiently studied.

A person’s attachment dimension has further been suggested to influence one’s decision to take on psychotherapy or not when facing psychological distress [17]. Furthermore, a commonly discussed reason for lower psychotherapy use in men as compared to women is the endorsement of traditional masculinity ideology with the two main foci of “be in control” and “be unlike women” [13,14,18,19,20]. It has been shown in several studies that the endorsement of traditional masculinity ideology is significantly associated with attachment orientation, suggesting a shared potential to explain psychotherapy use [21,22,23,24]. Regardless of one’s endorsement of traditional masculinity ideology, a person’s attachment orientation appears to be critical to the engagement in psychotherapy, as it has a central role in approaching and acting in social relationships and self-disclosure [17]. Attachment orientation is operationalized using a two-dimensional conceptualization, namely of attachment anxiety and attachment avoidance [25,26]. Anxiously attached individuals generally have a negative view of the self, tend to be dependent on others, and are hypervigilant to social and emotional cues from others [27,28,29]. By contrast, individuals exhibiting an avoidant attachment orientation generally perceive others as unavailable, unresponsive, or punitive [25,29]. Thus, individuals with an avoidant attachment orientation engage more in deactivation strategies by denying the importance of relationships and avoiding emotional intimacy [30]. Therefore, individuals exhibiting low attachment anxiety and avoidance generally view themselves as valuable and perceive others as trustworthy [25,29].

Many individuals who seek psychotherapeutic services exhibit anxious or avoidant attachment orientations, while in psychotherapy, the client–therapist relationship has many characteristics of an attachment relationship [31]. The therapist is supposed to serves as a secure base, acting as a potential agent of change to support the client in developing his intimate connections, social relationships, and autonomous exploration. A secure attachment organization with the therapist was shown to predict positive working alliance, more compliance, and greater engagement in therapy [17,32,33]. By contrast, stronger attachment avoidance was associated with rejection of treatment providers, reduced self-disclosure, and poorer use of treatment [17], which are rather typical issues in male clients [12,34], highlighting the question whether gender differences in attachment anxiety and avoidance are associated with psychotherapy use.

A significant problem in research on attachment theory is rooted in the fact that different research traditions, each with different questions and measurement methods, lead to different results. For example, the categorical approach of attachment research can be attributed more to a developmental psychological research approach in which observer assessment procedures, such as an interview or projective procedures, predominate. The dimensional research approach can instead be attributed to the field of personality and social psychology, in which questionnaires (self-assessment procedures) represent the method of choice [35,36]. This is also evident in the question of gender differences. Gender differences in attachment organization have been reported early on, with men being overrepresented in the attachment avoidant category as compared to women when using the adult attachment interview (AAI) [17]. Yet, a meta-analysis examining gender differences in the AAI did not identify consistent gender differences in attachment representations [37]. This is contrasted by consistently reported gender differences when using a dimensional approach with self-report questionnaires to assess attachment dimensions [38,39,40]. A meta-analysis examining attachment dimensions measured with psychometric scales (Experience in Close Relationships [ECR]; Experience in Close Relationships-Revised [ECR-R]; Adult Attachment Questionnaire) reveals that women exhibit higher attachment anxiety, while men exhibit higher attachment avoidance [41]. This further led to the claim to consistently integrate gender differences between men and women in attachment research [42,43].

Potential gender differences in attachment organization in university students are still insufficiently examined and might help to further explain gender differences in psychotherapy use in this population and ultimately inform tailored mental health support systems for male and female students. To date, no study has investigated whether men and women differ in their choice to take on psychotherapy depending on specific psychological syndromes and their level in attachment anxiety or avoidance. Based on the outlined literature on gender differences in psychotherapy use and attachment organization, we hypothesize that individuals with higher attachment anxiety, having a more negative view of the self, and being more dependent on others will use psychotherapy more than individuals with low attachment anxiety. For attachment avoidance, the empirical basis leads less unambiguously to distinct hypotheses. Here, two scenarios would be possible. On the one hand, for individuals with high attachment avoidance, who generally perceive others as unavailable, unresponsive, or punitive and deny the importance of relationship and avoid emotional closeness, it can be assumed that these individuals are reluctant to engage in psychotherapy. However, on the other hand, it can be assumed that with high attachment avoidance, major interpersonal problems arise in the life course and that these persons then finally make use of psychotherapy. Therefore, it can be assumed that attachment avoidance is also positively associated with the use of psychotherapy, but that this is less pronounced than in the case of attachment anxiety. Since women are expected to have higher attachment anxiety and men higher attachment avoidance, these gender differences in attachment organization may partly explain gender differences in psychotherapy use.

## 2. Materials and Methods

### 2.1. Sample and Procedure

The study was conducted at the University of Cologne in the winter semester 2014/2015 (December 2014 to February 2015) as part of an online survey (KUmBel; Kölner Umfrage bei Studierenden zu psychischen Belastungen). A description of the study sample and survey procedure are reported in detail elsewhere [44]. The University of Cologne is divided into six faculties: the Faculty of Economics and Social Sciences, the Faculty of Law, the Faculty of Medicine, the Faculty of Philosophy, the Faculty of Mathematics and Natural Sciences, and the Faculty of Human Sciences.

Of the 49,772 students enrolled in the winter semester 2014/2015, a total of 44,299 students at the University of Cologne were invited to participate in the online survey via the uniform e-mail distribution list. This is because during the time the survey invitation was sent via the uniform e-mail distribution system of the University of Cologne, the e-mail system of the Mathematical and Natural Sciences Faculty was not yet integrated into the uniform e-mail distribution system, and thus their members were not reached. These technical hurdles led to the inability to contact 5473 students. Survey exclusion criteria were, therefore, (i) not being enrolled as a student at the University of Cologne at time of survey, (ii) not having a student account provided by the University of Cologne at the time of survey, and (iii) being a member of the Mathematical and Natural Sciences Faculty of the University of Cologne. For a detailed sample flow leading to the final sample, see Appendix A.

With the invitation email, students received a link to an online platform [45], where the survey could be completed. After four weeks, a reminder email was sent out to potential participants. For survey completion, participants needed about 20–30 min. The Ethics Committee of the Medical Faculty of the University of Cologne gave its positive vote after extensive consultation on the issues relevant to data protection.

### 2.2. Measures

Participants were asked several sociodemographic questions (e.g., age, gender, nationality, previous vocational training) and course-related questions (e.g., course of study, subject semesters, ways of financing studies, thoughts of dropping out) were collected. Gender was assessed using a binary response option, where students could indicate to self-identify as “woman” or “man”. In addition, information on current and intended psychotherapy use for mental health problems was obtained. Regarding current psychotherapy use, students were asked directly whether they were currently receiving psychotherapy (“Yes”/“No”), while the question about intention to start psychotherapy asked students whether they had thought about receiving psychotherapy (“Yes”/“No”). In a previous study with German-speaking students, the item on current psychotherapy use showed high reliability (Cohen’s kappa κ = 0.92) over a one-month period, as well as significant convergent validity with depression symptoms (PHQ-9; point biserial correlation coefficient *r*_pb_ = 0.18, *p* < 0.001) and anxiety symptoms (GAD-7; *r*_pb_ = 0.17, *p* < 0.001) [46]. On the other hand, the same study supported the discriminant validity of this item with significant negative correlations between current psychotherapy use and self-esteem (*r*_pb_ = −0.19, *p* < 0.001), optimism (*r*_pb_ = −0.10, *p* = 0.009), and resilience (*r*_pb_ = −0.11, *p* = 0.008) [46]. In the present study, convergent validity of the two items was further supported by a significant positive correlation between current psychotherapy use and suicidal ideation (Pearson’s phi coefficient *r*_φ_ = 0.15, *p* < 0.001) as well as between the intention to start psychotherapy and suicidal ideation (*r*_φ_ = 0.30, *p* < 0.001).

The mental distress of the students was assessed with the Patient Health Questionnaire (PHQ-D; [47], the German version of the Patient Health Questionnaire (PHQ; [48]). Designed as a self-report instrument, the PHQ-D is a standard instrument for screening the most common mental disorders. In addition to psychosocial functioning and eight common psychosocial stressors, the complete version (78 items) covers a total of eight psychological disorders according to DSM-IV criteria. A previous study reported an internal consistency for the depression module of α = 0.88 and for the somatization module of α = 0.79 [49]. In the present study, comparable values for the internal consistency were identified for the depression module of α = 0.86 and for the somatization module of α = 0.76.

We used the Experiences in Close Relationship-Revised (ECR-RD12) questionnaire to measure attachment anxiety and attachment avoidance [50,51]. The ECR-RD12 is used to measure partnership-related attachment, as in the sense of attachment in close but not exclusively romantic relationships, such as with romantic partners, but also with parents or close friends. The 12-item scale measures the two dimensions of attachment: anxiety (6 items) and avoidance (6 items). Sample items are “I worry quite a bit about losing connection with other people” (anxiety) and “I don’t feel comfortable opening up to other people” (avoidance). Responses were given on a 7-point Likert scale from “strongly disagree” to “strongly agree”. Previously, high internal consistency was reported for the German translation of the ECR-R with Cronbach’s α = 0.91 and 0.92 for the two scales. In the present study, the two scales revealed an internal consistency of α = 0.82 for attachment anxiety and for attachment avoidance of α = 0.78.

### 2.3. Statistical Analysis

The statistical analysis was conducted in the R software environment [52], including the additional R-packages “psych” ([53]; calculation of internal consistencies), “car” ([54]; calculation of variance inflation factors), “rcompanion” ([55]; maximum likelihood estimates of pseudo *R*^2^ and *p*-values in the logistic regression setting), and “ggplot2” ([56]; visualizations). The analysis consisted of the four steps described in the following, with the covariates age and nationality included in all calculations. To test significance of the results, an initial alpha-level of α = 0.05 was used for each step. Subsequently, the results were checked for robustness using the Holm–Bonferroni correction for multiple testing. Furthermore, for all calculations where the ECR-RD12 was involved, a slightly reduced sample of *N* = 4705 was used due to some missing values.

In the first step, the sample characteristics were analyzed by calculating mean scores for the continuous variables and relative percentages for the categorical variables, once for the total sample and once for men and women separately. In the second step, group differences between men and women with regard to attachment anxiety and attachment avoidance, screened psychological syndromes, and psychotherapy use were analyzed using two-sided *t*-tests for the continuous variables and two-sided Wald tests for the categorical variables. In the third step, binary logistic regression models were fitted to assess the predictive value of attachment orientation for the likelihood to use or the intention to start psychotherapy, once in the total sample, once for men and women separately, and once for all individual subgroups consisting of participants screened positive for a psychological syndrome. In the fourth and last step, moderation analyses were conducted using Hayes’ [57] regression-based approach to test for possible moderating effects of attachment orientation on the association between the likelihood to use or the intention to start psychotherapy and the positive screening for different psychological syndromes. 

Assumptions for the previously described analyses and statistical tests were assessed by using Levene’s median-based test for equal variances [58]; testing homoscedasticity for the *t*-tests, Cook’s distance [59]; detecting influential points in the logistic regression models, and the generalized variance inflation factor proposed by Fox and Monette [60]; detecting collinearity of predictor variables in the logistic regression models.

## 3. Results

### 3.1. Sample Characteristics

As presented in Table 1, the total sample of the study consisted of 4894 participants, of which 1207 self-identified as men (24.7%) and 3687 self-identified as women (75.3%). The vast majority were of German nationality (94.7%). About one-fifth of the students had previously completed another degree program (19.4%), and while about one-quarter had previously dropped out of another degree program (24.2%), nearly half of the students were currently thinking about dropping out of their current program (45.6%). The majority of participants received financial support either from their relatives (70.5%) or through an additional job alongside their studies (67.8%). Though most of the students previously had contact with a counseling service (71.2%) and about one-quarter previously used psychotherapeutic treatment (26.0%), fewer than every tenth student was currently using psychotherapy (8.7%). Nonetheless, about every fourth student expressed the intention to use psychotherapy in the future (28.4%). Furthermore, about one-fifth of the students previously required medical care (21.4%) and a small minority was currently undergoing medical treatment (6.9%). Additionally, about one-tenth of the students reported suffering from a physical impairment (9.4%), about one-twentieth was currently taking neuropsychopharmacological medication (6.0%), and a minority previously experienced physical or sexual abuse (3.5%).

Regarding psychological distress, more than half of the students screened positive for at least one psychological syndrome (58.4%). Most students screened positive for a depressive syndrome (34.9%), followed by a somatoform syndrome (23.6%), and an alcohol use syndrome (19.1%). Roughly the same number of students suffered from an anxiety syndrome (12.5%) as suffered from an eating disorder syndrome (11.8%).

### 3.2. Group Comparisons

As presented in Figure 1 and Table 1, women had a significantly higher mean score for attachment anxiety compared to men, while men had a significantly higher mean score for attachment avoidance as compared to women. These differences also remained significant after applying the Holm–Bonferroni correction for multiple testing.

In terms of positive psychological syndrome screenings with the PHQ-D (Figure 2, Table 1), there was no statistically significant difference between men and women in the aggregate category of any syndrome, while significantly more women than men were screened positive for any depressive syndrome (minor and/or major depression), any anxiety syndrome (panic and/or generalized anxiety), and generalized anxiety syndrome, and while significantly more men than women were screened positive for alcohol use syndrome, significantly more women than men were screened positive for a somatoform syndrome and any eating disorder syndrome (bulimia nervosa and/or binge eating). However, after applying the Holm–Bonferroni correction for multiple testing, the gender differences for any depressive syndrome and any eating disorder syndrome did no longer reach statistical significance. Regarding psychotherapy use, men were about 26.9% less likely to use psychotherapy compared to women and men were also about 23.8% less likely to have the intention to start psychotherapy in the future compared to women. These differences remained significant after a Holm–Bonferroni correction.

Analyzing individual subgroups screened positive for a psychological syndrome with regard to gender differences in current psychotherapy use (Figure 3A), it emerged that men screened positive for at least one psychological syndrome (9.0%) were significantly less likely to currently use psychotherapy compared to women screened positive for at least one psychological syndrome (11.5%; *z* [2851] = −2.33, *p* = 0.020, Nagelkerke *R*^2^ = 0.022, AIC = 1947.9). However, this result was no longer significant after applying the Holm–Bonferroni correction. Examining the same subgroups for gender differences in the intention to start psychotherapy (Figure 3B) revealed that significantly fewer men screened positive for any syndrome (32.0%) were intending to start psychotherapy in the future compared to women screened positive for any syndrome (39.9%; *z* [3777] = −3.72, *p* ≤ 0.001, Nagelkerke *R*^2^ = 0.008, AIC = 3787). Similarly, fewer men screened positive for major depression (48.3%), alcohol use syndrome (27.6%), and any eating disorder syndrome (28.5%) were intending to start psychotherapeutic treatment in the future compared to women screened positive for major depression (57.7%), alcohol use syndrome (36.0%), and any eating disorder syndrome (47.8%), (major depression: *z* [1002] = −2.42, *p* = 0.015, Nagelkerke *R*^2^ = 0.010, AIC = 1385.9; alcohol use syndrome: *z* [929] = −2.77, *p* = 0.006, Nagelkerke *R*^2^ = 0.014, AIC = 1185; any eating syndrome: *z* [751] = −2.34, *p* = 0.019, Nagelkerke *R*^2^ = 0.023, AIC = 763.3). After correcting these findings for multiple testing, the only difference that remained significant was the gender difference in the subgroup screened positive for any psychological syndrome.

### 3.3. Logistic Regressions

Fitting binary logistic regression models to predict current psychotherapy use based on attachment orientation in different subgroups (Figure 4, Appendix A), showed that higher attachment anxiety predicted higher odds to currently use psychotherapy in the total sample (Figure 4A), as well as in all other examined subgroups (Figure 4B–I). These results remained significant even after applying a Holm–Bonferroni correction. On the other hand, higher attachment avoidance did predict higher odds for current psychotherapy use in male students (Figure 4B) and in students screened positive for alcohol use disorder (Figure 4G). However, these results for attachment avoidance became nonsignificant after applying the Holm–Bonferroni correction for multiple testing. Similarly, attachment avoidance did not have any predictive value for current psychotherapy use for the total sample (Figure 4A), for female students (Figure 4C), or any of the other subgroups screened positive for a psychological syndrome (Figure 4D–F,H,I).

Examining the same subgroups, but predicting the intention to start psychotherapy as an outcome based on attachment orientation (Figure 5, Appendix A), showed that higher attachment anxiety was a significant predictor for higher odds of intending to start psychotherapy in the total sample (Figure 5A), as well as in all other subgroups that were analyzed (Figure 5A–I). Again, these results remained significant after correcting for multiple testing. Similarly, higher attachment avoidance would predict higher odds of intending to start psychotherapy in the total sample (Figure 5A) and in all subgroups except in students screened positive for any eating disorder syndrome (Figure 5I). Yet, after applying the Holm–Bonferroni correction, attachment avoidance was only significantly associated with the intention to start psychotherapy in the total sample (Figure 5A), in female students (Figure 5B), in students screened positive for any psychiatric syndrome (Figure 5D), and in students screened positive for any depressive syndrome (Figure 5E). Further analyses, where the positively screened subgroups were additionally separated by gender, can be found in the Appendix A.

As purely exploratory analyses reported in the Appendix A, moderation analyses were carried out investigating the possible moderating effects of attachment orientation on the association between individual psychiatric syndromes and psychotherapy use (see Appendix A).

## 4. Discussion

### 4.1. Summary of Results

In the present study, a significant gender difference was observed in the two attachment dimensions of attachment anxiety and attachment avoidance. Male university students showed significantly lower attachment anxiety and significantly higher attachment avoidance. Furthermore, male students reported to engage significantly less in psychotherapy as compared to female students, and also reported lower intentions to use psychotherapy in the near future as female students. Male and female students did not differ in general psychological distress, while male students tend to show lower rates in positive syndrome screenings with the PHQ-D for all syndromes except for the alcohol use disorder syndrome, where for far more male students than female students a positive screening was observed. In male students, current psychotherapy use was predicted by attachment anxiety and avoidance, while only the association between attachment anxiety and psychotherapy use survived correction for multiple testing. In female students only attachment anxiety emerged as a significant predictor of psychotherapy use. In exploratory analyses (see Appendix A), attachment anxiety emerged as a significant moderator of the association between suffering from a depressive or somatoform syndrome and current psychotherapy use in female, but not male, students, with higher attachment anxiety leading to increased psychotherapy use. However, moderation effects did not remain significant after correction for multiple testing.

### 4.2. Integration of Findings

As widely established [61,62,63], also in the present study men and women and more precisely, male students and female students, did not differ in overall psychological distress as measured by the PHQ-D any syndrome. Although there are biologically based approaches explaining gender differences in the prevalence rates of specific psychological disorders [64,65], these approaches, while highlighting gender specific treatment options, can only explain gender differences in these disorders to a certain extent [66,67,68]. While the often-reported elevated levels in psychological distress in women [69] seems better explained by differing gender role norms in men and women leading to differing distress reporting and symptom presentation [70], but not the overall prevalence in psychological disorders [11]. However, as presented in Figure 2 and in accord with large multi-national investigations and meta-analyses, male students showed a reduced rate of any anxiety syndrome, general anxiety syndrome, and somatoform syndrome as compared to female students, while the alcohol use syndrome was the only syndrome for which male students showed significantly higher rates than female students [1,2,3,4,5,7,71]. Lower rates for male students in any depression syndrome or any eating syndrome did not remain statistically significant after correction for multiple testing.

With regard to psychotherapy use, male students showed a significantly lower current psychotherapy use and also a lower intention to use psychotherapy in the near future as compared to female students (see Table 1, Figure 2), which further corroborates previous reports from Germany or Switzerland [14,15,16]. However, when investigating gender differences in psychotherapy use by syndrome, only for the category any syndrome a significant difference was initially detected, which did not remain significant after correction for multiple testing. While for panic disorder syndrome, proportionally more male students use psychotherapy than female students, for all other syndromes, female students use psychotherapy proportionally more (see Figure 3A). Examining the intention to use psychotherapy in those affected by any syndrome, male students reported significantly lower intention to take on psychotherapy. This, however, was also detected on syndrome level for major depressive syndrome, alcohol use syndrome, and any eating syndrome. Yet, after correcting for multiple testing only the category any syndrome remained statistically significant (see Figure 3B). As there are no gender-specific comparisons with regard to current or intended psychotherapy use by syndrome, these results provide important new insights for future research on gender differences in psychotherapy use for specific disorders.

In order to answer the question about the predictive potential of attachment dimensions with regard to psychotherapy use, it is first necessary to clarify whether there are also gender differences in attachment dimensions in the present study. Confirming previous findings [42], gender differences in attachment dimensions were corroborated (see Figure 1). Gender differences in attachment organization have been reported already thirty years ago [17]. However, due to imprecise use of attachment terminology and two different research backgrounds (developmental psychology [AAI] and personality psychology [ECR]), there is much confusion with respect to gender differences in attachment research. These two research approaches to quantifying attachment are entirely different methods, which is why the outcomes obtained cannot be compared in terms of gender differences in attachment organization in general. Nevertheless, when assessing attachment dimensions with self-report measures, men generally exhibit lower attachment anxiety and higher attachment avoidance [41]. Our study corroborates these results and highlights gender differences in attachment orientation as potential explanatory factor for lower psychotherapy use in men. Furthermore, it seems arguably necessary to additionally examine gender differences in attachment dimensions at the subgroup level and in a culture-specific manner [41,72], with university students from Germany representing one such subgroup [44].

As shown in Figure 4, for the male and female sample, attachment anxiety emerged as a significant predictor of current psychotherapy use. In the male sample, attachment avoidance also emerged initially as a significant predictor, but after correcting for multiple testing, the association faded. However, for attachment anxiety, a generally larger effect emerged suggesting attachment anxiety to be the more relevant behavior-regulating construct with regard to psychotherapy use. However, with regard to intention to use psychotherapy, in the total and the female sample, attachment anxiety and avoidance significantly predicted intended psychotherapy use, while in the male sample only attachment anxiety was a significant predictor after correcting for multiple testing (see Figure 5). Again, attachment anxiety seems to be the more behavior-regulating construct with respect to intended psychotherapy use with consistently higher odds ratios; although the difference in effect size became less clear especially in the female subsample. When examining gender differences in the relationship between attachment organization and psychotherapy use (current and intended), it is striking that a similar association appears to exist for all syndromes examined. However, it must be added here that due to power reduction in these specific subsample analyses, several associations no longer became statistically significant, especially for the male sample by specific syndromes with regard to attachment avoidance (see Appendix A).

Many people seek psychotherapeutic support because of interpersonal problems caused by high attachment anxiety or avoidance. Insecure attachment operationalized as either high attachment anxiety or avoidance has consistently been associated with various psychological disorders [37,73]. There are several studies further showing an association between insecure attachment and poor affect regulation [74], worse social skills [75], more negative self-evaluations [76], and worse problem-solving competencies [77]. Therefore, it follows theoretically that these two constructs are positively associated with psychotherapy utilization. However, to date, this has never been investigated and reported in a gender separate manner. Thus, this is the first study to show psychotherapy use to be positively associated with attachment anxiety and avoidance in a gender-specific manner.

In psychotherapy, the client–therapist relationship resembles an attachment relationship [31]. Therefore, the finding that attachment avoidance is less strongly associated with psychotherapy use than attachment anxiety is plausible. Avoidantly attached individuals expect relevant others to be unavailable, unresponsive, or punitive [25,33]. This causes insecurities for the potential psychotherapy clients, which are either overcome with courage and willpower, or are, at a certain point, no longer relevant enough due to the increasing psychological distress. This leads to the question of whether individuals with predominantly avoidant attachment issues, compared to individuals with predominantly anxious attachment issues, tend to enter psychotherapy at later stages of a disease course and exhibit correspondingly higher levels of distress. Indicative of this is that stronger attachment avoidance was associated with rejection of treatment providers, reduced self-disclosure, and poorer use of treatment [17], while more securely attached individuals seem to benefit more from psychotherapy [78]. A secure attachment orientation with the therapist is associated with a better working alliance, more compliance, and greater engagement in therapy [17,32,33].

Anxiously attached individuals, however, tend to depend on others for help with uncertainties and, therefore, tend to consider psychotherapy even when experiencing only minor psychological symptoms. In addition, anxiously attached people often have a negative view of themselves, so that they are generally more insecure whether they are “good” and “okay” as they are and might secretly wish to address this insecurity in psychotherapy when psychological symptoms arise [33]. This line of argumentation fits perfectly with the observed effect size difference in the predictive effect of the two attachment orientations on psychotherapy use with attachment anxiety being the more behavior-regulating construct.

Finally, we investigated in an explorative manner potential moderating effects of attachment anxiety and avoidance on the relation between suffering from a mental health syndrome and psychotherapy use (Appendix A). In the total sample, initially significant moderation effects were detected indicating that attachment anxiety moderated the association between any depressive syndrome and current psychotherapy use with higher attachment anxiety levels being associated with an increased likelihood of using psychotherapy. A similar effect was identified for the relation between somatoform syndrome and current psychotherapy use moderated by attachment anxiety. However, the effects did not hold after correction for multiple testing. For the outcome intended psychotherapy, only in the female sample did attachment avoidance emerge as a significant moderator in combination with any depressive syndrome as predictor, which also did not remain significant after correction for multiple testing. In summary, with regard to the moderation analyses, it can be concluded that the findings provide an inconsistent picture, small effect size results, and, therefore, should be interpreted with caution.

### 4.3. Limitations

There are several limitations of the study when interpreting the results. First, the cross-sectional nature of the study does not allow any conclusions about causality and large longitudinal designs are required to examine time-dependent associations. Second, gender was assessed only in a binary manner (Woman/Man). Although this was regular at the time the study was conducted, future studies would need to further separate the gender variable into the following categories: woman, man, transgender, cisgender, non-binary, other, or people preferring not to answer this question. Third, although the PHQ-D is a valid and reliable instrument for identifying mental disorders at the syndrome level, a clinical diagnostic interview would be preferable. Due to the lack of differential diagnostics in this instrument, it is not possible to determine conclusively whether, for example, a person with a positive screen for “any depression” is not primarily suffering from another mental disorder, e.g., anxiety disorders or substance use disorders, or even from a somatic disease, e.g., hypothyroidism. Fourth, the study was specifically mentioning mental stress in its title, potentially biasing participation towards a convenience sample with comparably more mental health problems. Fifth, although widely discussed in the context of psychotherapy reluctance, the present study did not include a measure for traditional masculinity. Sixth, in terms of survey completion, 189 individuals did not complete the ECR-RD12. If it were assumed that these dropouts did not occur at random, they could possibly differ in a systematic way in attachment anxiety and avoidance due to the underlying group membership. However, the sensitivity analysis showed no differences in the results when only the reduced sample was used.

### 4.4. Conclusions

Lower attachment anxiety among male students compared with female students provides new insights with regard to lower psychotherapy use among mentally distressed male students. Since anxiously attached individuals suffer from a more negative view of the self, rendering them more prone to hyperactivating strategies such as being overdependent on others and more often seeking reassurance or guidance, female students exhibiting overall higher levels in attachment anxiety use psychotherapy more often than men also because of this reason. As shown by logistic regressions, overall attachment anxiety is more behavior regulating than attachment avoidance with regard to psychotherapy use. Avoidantly attached individuals generally perceive others as unavailable, unresponsive, or punitive, which is associated with more deactivating strategies (e.g., avoiding emotional intimacy, denial of importance of emotional relationships). Therefore, higher attachment avoidance in male students cannot balance psychotherapy use between male and female students due to significantly higher attachment anxiety in female students and its stronger association with psychotherapy use. Public health campaigns to increase psychotherapy use among men should integrate these findings, and general practitioners should be aware of this and support attachment avoidant men with mental health issues to overcome reluctance toward psychotherapy. Likewise, psychotherapy practitioners should consider and address their male clients’ attachment organization due to an increased risk of psychotherapy dropout in avoidantly attached men.

## Figures and Tables

**Figure 1 behavsci-12-00204-f001:**
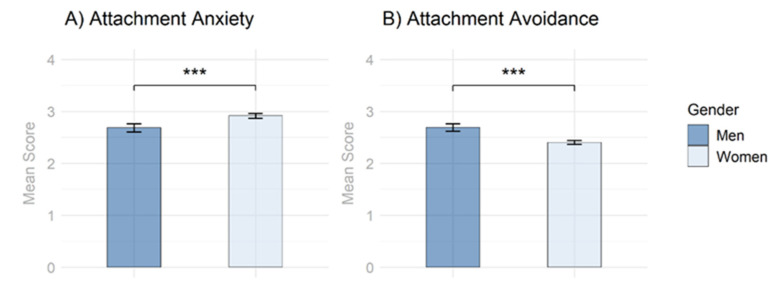
Gender Differences Regarding Attachment Dimensions. (**A**) Gender Differences Regarding Attachment Anxiety. (**B**) Gender Differences Regarding Attachment Avoidance. Note. *p*-values were adjusted for multiple testing using the Holm–Bonferroni method. *** = *p* < 0.001.

**Figure 2 behavsci-12-00204-f002:**
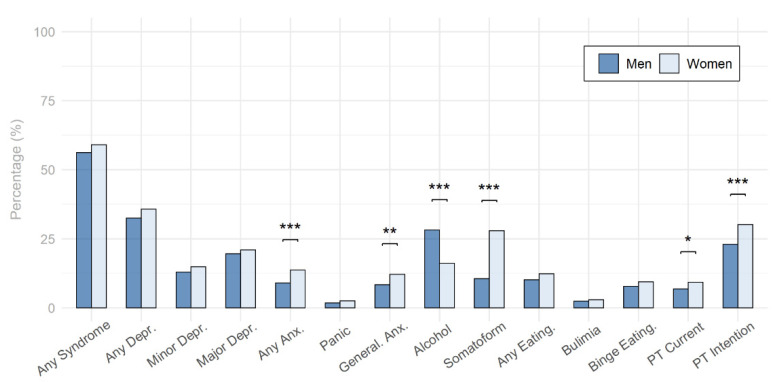
Gender Differences Regarding Screened Psychological Syndromes and Psychotherapy Use. Note. Depr. = Depression, Anx. = Anxiety, General. = Generalized, Eating. = Eating disorder, PT = Psychotherapy. *p*-values were adjusted for multiple testing using the Holm–Bonferroni method. * = *p* < 0.05, ** = *p* < 0.01, *** = *p* < 0.001.

**Figure 3 behavsci-12-00204-f003:**
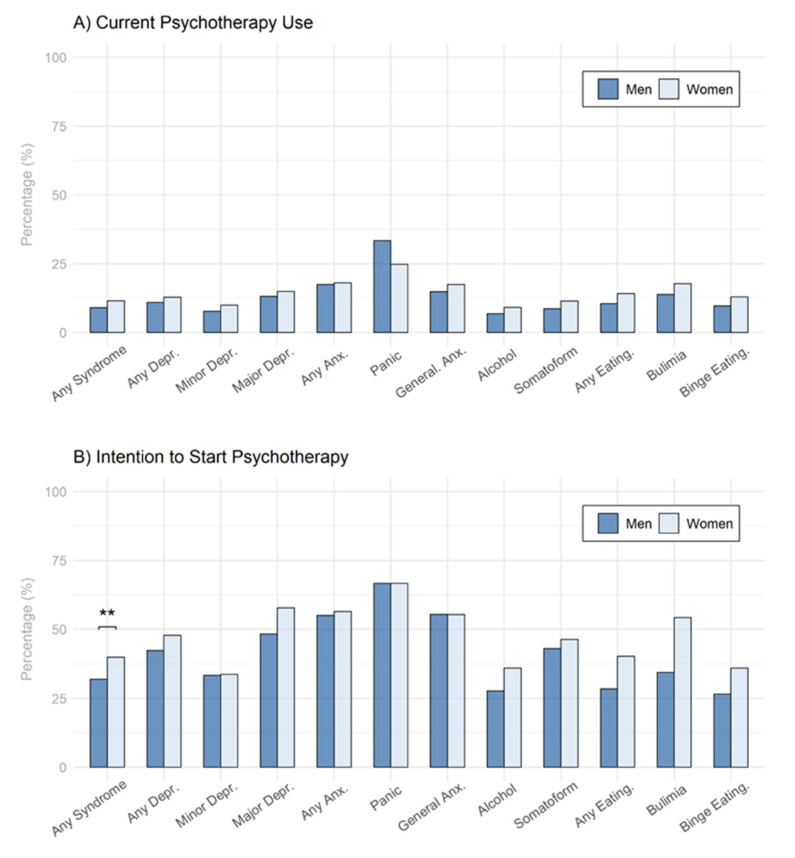
Gender Differences Regarding Psychotherapy Use in different Syndromic Subgroups. (**A**) Gender Differences Regarding Current Psychotherapy Use. (**B**) Gender Differences Regarding Intention to Start Psychotherapy. Note. Depr. = Depression, Anx. = Anxiety, General. = Generalized, Eating. = Eating disorder, PT = Psychotherapy. *p*-values were adjusted for multiple testing using the Holm–Bonferroni method. ** = *p* < 0.01.

**Figure 4 behavsci-12-00204-f004:**
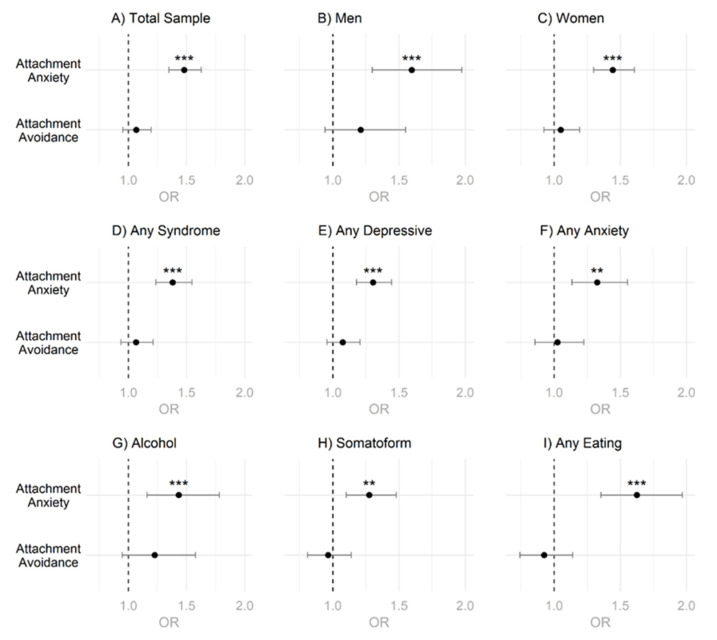
Odds Ratios for Attachment Dimensions and Current Psychotherapy Use. (**A**) Analysis for Total Sample. (**B**) Analysis for Men. (**C**) Analysis for Women. (**D**–**I**) Analysis for Specific Syndromic Groups. Note. OR = Odds Ratio. *p*-values were adjusted for multiple testing using the Holm–Bonferroni method. ** = *p* < 0.01, *** = *p* < 0.001.

**Figure 5 behavsci-12-00204-f005:**
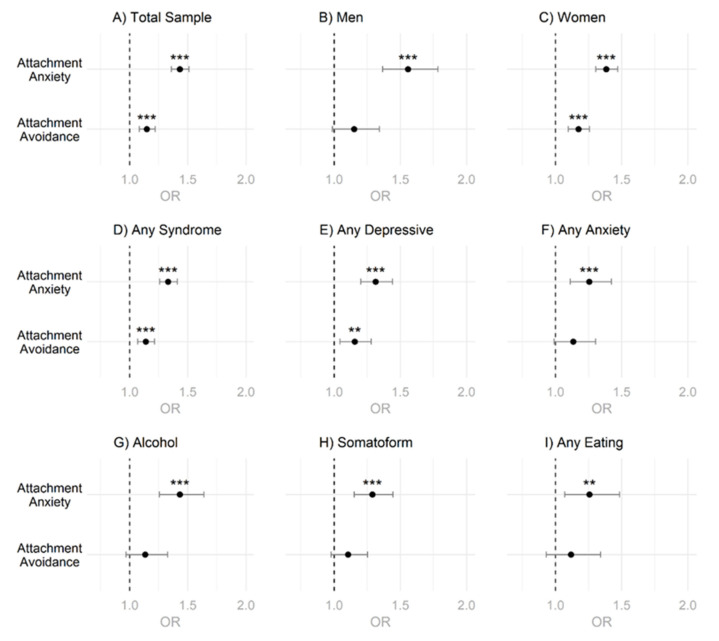
Odds Ratios for Attachment Dimensions and the Intention to Start Psychotherapy. (**A**) Analysis for Total Sample. (**B**) Analysis for Men. (**C**) Analysis for Women. (**D**–**I**) Analysis for Specific Syndromic Groups. Note. OR = Odds Ratio. *p*-values were adjusted for multiple testing using the Holm–Bonferroni method. ** = *p* < 0.01, *** = *p* < 0.001.

**Table 1 behavsci-12-00204-t001:** Descriptive Statistics for the Sample.

	Total (*N* = 4894)		Men (*N* = 1207)		Women (*N* = 3687)				
	*N (%)*	*M (SD)*	*N (%)*	*M (SD)*	*N (%)*	*M (SD)*	Test Statistic *(**df)*	*p*	*p (corr.)*
**Age**		24.3 *(4.9)*		25.1 *(5.2)*		24.1 *(4.8)*	6.43 *(4890)*	**<0.001 *****	**<0.001 *****
**Nationality**									
German	4637 *(94.7)*		1150 *(95.3)*		3487 *(94.6)*		1.55 *(4890)*	0.122	0.224
Non-German	339 *(6.9)*		82 *(6.8)*		257 *(7.0)*		1.22 *(4890)*	0.222	0.224
**Study situation**									
Previous degree	950 *(19.4)*		226 *(18.7)*		724 *(19.6)*		−2.69 *(4889)*	**0.007 ****	**0.021 ***
Previous dropout	1182 *(24.2)*		324 *(26.8)*		858 *(23.3)*		1.44 *(4889)*	0.149	0.149
Thinking about dropout	2233 *(45.6)*		583 *(48.3)*		1650 *(44.8)*		1.87 *(4889)*	0.062	0.124
**Financed by**									
Relatives	3450 *(70.5)*		845 *(70.0)*		2605 *(70.7)*		1.25 *(4889)*	0.211	0.422
Job	3319 *(67.8)*		814 *(67.4)*		2505 *(67.9)*		−1.49 *(4889)*	0.137	0.411
BAföG	1185 *(24.2)*		244 *(20.2)*		941 *(25.5)*		−3.37 *(4889)*	**<0.001 *****	**0.004 ****
Scholarship	217 *(4.4)*		68 *(5.6)*		149 *(4.0)*		2.58 *(4889)*	**0.001 ****	**0.039 ***
Other means	456 *(9.3)*		123 *(10.2)*		333 *(9.0)*		0.15 *(4889)*	0.882	0.882
**Counseling/Therapy**									
Contact with counseling service	3485 *(71.2)*		895 *(74.2)*		2590 *(70.2)*		3.41 *(4889)*	**<0.001 *****	**0.001 ****
Previous psychotherapy	1273 *(26.0)*		272 *(22.5)*		1001 *(27.1)*		−4.50 *(4889)*	**<0.001 *****	**<0.001 *****
Current psychotherapy	425 *(8.7)*		83 *(6.9)*		342 *(9.3)*		3.06 *(4889)*	**0.002 ****	**0.002 ****
Intention psychotherapy	1388 *(28.4)*		278 *(23.0)*		1110 *(30.1)*		4.79 *(4889)*	**<0.001 *****	**<0.001 *****
**Medical Conditions**									
Previous medical care	1049 *(21.4)*		218 *(18.1)*		831 *(22.5)*		4.09 *(4889)*	**<0.001 *****	**<0.001 *****
Current medical care	337 *(6.9)*		64 *(5.3)*		273 *(7.4)*		2.91 *(4889)*	**0.004 ****	**0.016 ***
Physical impairment	459 *(9.4)*		103 *(8.5)*		356 *(9.7)*		1.67 *(4881)*	0.095	0.117
Physical / sexual abuse	173 *(3.5)*		53 *(4.4)*		120 *(3.3)*		1.93 *(4770)*	0.053	0.117
Medication	295 *(6.0)*		60 *(5.0)*		235 *(6.4)*		−2.07 *(4770)*	**0.039 ***	0.117
**Psychological Syndromes**									
Any syndrome	2856 *(58.4)*		678 *(56.2)*		2178 *(59.1)*		−1.72 *(4889)*	0.085	0.399
Any depressive	1710 *(34.9)*		392 *(32.5)*		1318 *(35.7)*		−2.05 *(4887)*	**0.040 ***	0.322
Minor depression	703 *(14.4)*		156 *(12.9)*		547 *(14.8)*		−1.51 *(4887)*	0.130	0.399
Major depression	1007 *(20.6)*		236 *(19.6)*		771 *(20.9)*		−1.10 *(4887)*	0.271	0.542
Any anxiety	614 *(12.5)*		109 *(9.0)*		505 *(13.7)*		−4.51 *(4889)*	**<0.001 *****	**<0.001 *****
Anxiety–panic	114 *(2.3)*		21 *(1.7)*		93 *(2.5)*		−1.75 *(4889)*	0.080	0.3985
Anxiety–generalized	549 *(11.2)*		101 *(8.4)*		448 *(12.2)*		−3.82 *(4889)*	**<0.001 *****	**0.001 ****
Alcohol use syndrome	934 *(19.1)*		340 *(28.2)*		594 *(16.1)*		−9.52 *(4889)*	**<0.001 *****	**<0.001 *****
Somatoform syndrome	1157 *(23.6)*		128 *(10.6)*		1029 *(27.9)*		−11.71 *(4889)*	**<0.001 *****	**<0.001 *****
Any eating disorder	576 *(11.8)*		123 *(10.2)*		453 *(12.3)*		−2.01 *(4889)*	**0.044 ***	0.322
Eating–bulimia	136 *(2.8)*		29 *(2.4)*		107 *(2.9)*		−0.85 *(4541)*	0.398	0.542
Eating–binge	441 *(9.0)*		94 *(7.8)*		347 *(9.4)*		−1.84 *(4889)*	0.065	0.391
**ECR-RD12 ^a^**									
Attachment anxiety		2.86 *(1.41)*		2.69 *(1.39)*		2.92 *(1.41)*	−4.80 *(4700)*	**<0.001 *****	**<0.001 *****
Attachment avoidance		2.47 *(1.23)*		2.69 *(1.25)*		2.40 *(1.22)*	6.88 *(4700)*	**<0.001 *****	**<0.001 *****

Note. *N* = number of participants, *M* = mean, *SD* = standard deviation, *df* = degrees of freedom, *p* = *p*-value, *test statistic* = *t*-value for continuous and *z*-score for categorical variables. *corr.* = Holm–Bonferroni adjustment for multiple testing. ^a^ ECR-RD12 = Experiences in Close Relationship–Revised.* = *p* < 0.05, ** = *p* < 0.01, *** = *p* < 0.001.

## Data Availability

The data used for the present study will be made available by the corresponding author upon request.

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
