# Peer review of "Gender Differences in Attachment Anxiety and Avoidance and Their Association with Psychotherapy Use—Examining Students from a German University"

_behavsci, 2022, doi:10.3390/bs12070204_

Round 1

Reviewer 1 Report

The manuscript from Weber and colleagues found. The manuscript is well-written, and the experimental design sounds robust and supports the authors' results and conclusions. The authors correctly described their study limitations in a specific section in the discussion. I have only minor comments that could improve the manuscript:

11)     The manuscript would benefit from the description, in the methodology section, of the criteria used for eliminating a participant from the study.

22)     If possible, a copy of the questionaries used could be available in the supplementary material as it could help others interpret the results obtained in the study.

Author Response

We thank the reviewer for the careful evaluation of our manuscript. A response letter addressing the inquiries is provided.

Reviewer 2 Report

In the heading context of the use of attachment anxiety and avoidance need to be clear (e.g. romantic, non-romantic, general, etc.).

The literature review did not reflect the gender difference in the overall burden/prevalence of Psychiatric Syndromes mentioned in the result. The discussion may include these issues in adequate.  

is it possible to present a result in which each syndrome is accompanied with gender + psychotherapy. 

Also, make sure the word psychiatric syndrome appropriately represents your proposition or may be changed.

Author Response

(The authors gave the same response as above.)
